# Biperiden for prevention of post-traumatic epilepsy: A protocol of a double-blinded placebo-controlled randomized clinical trial (BIPERIDEN trial)

**Maira Licia Foresti**[1,2,3☯], **Eliana Garzon**[2,4☯], **Carla Cristina Gomes Pinheiro**[2], **Rafael Leite Pacheco**[5,6], **Rachel Riera**[6,7], **Luiz Eugênio Mello**[1,2,3]*

1 Neurology Neuroscience Postgraduation Program, Physiology Department, Universidade Federal de São Paulo, São Paulo, Brasil, 2 Sociedade Beneficente de Senhoras Hospital Sírio-Libanês, São Paulo, SP, Brazil, 3 Instituto D'Or de Pesquisa e Ensino, São Paulo, SP, Brasil, 4 Department of Neurology, Division of Neurologic Clinic, Hospital das Clínicas, Faculdade de Medicina, Universidade de São Paulo, São Paulo-SP, Brazil, 5 Departamento de Medicina, Centro Universitário São Camilo, São Paulo, SP, Brazil, 6 Center of Health Technology Assessment, Sociedade Beneficente de Senhoras Hospital Sírio-Libanês, São Paulo, SP, Brazil, 7 Discipline of Evidence-based Medicine, Escola Paulista de Medicina, Universidade Federal de São Paulo, São Paulo, SP, Brazil

☯ These authors contributed equally to this work.
* lemello@unifesp.br

**Data Availability Statement:** No datasets were generated or analyzed during the current study. All

## Abstract

### Background

Traumatic brain injury (TBI) is one of the most important causes of acquired structural epilepsy, post-traumatic epilepsy (PTE), however, efficient preventative measures and treatment are still not available to patients. Preclinical studies indicated biperiden, an anticholinergic drug, as a potential drug to modify the epileptogenic process. The main objective of this clinical trial is to evaluate the efficacy of biperiden as an antiepileptogenic agent in patients that suffered TBI.

### Methods

This prospective multicenter (n = 10) interventional study will include 312 adult patients admitted to emergency care units with a diagnosis of moderate or severe TBI. Following inclusion and exclusion criteria, patients will be randomized, using block randomization, to receive double-blind treatment with placebo or biperiden for 10 days. Follow-up will occur at specific time windows up to 2 years. Main outcomes are incidence of PTE after TBI and occurrence of severe adverse events. Other outcomes include exploratory investigation of factors that might have benefits for the treatment or might influence its results, such as genetic background, clinical progression, electroencephalographic abnormalities, health-related quality of life and neuropsychological status. Analyses will be conducted following the safety, intention-to-treat and efficacy concepts.

relevant data from this study will be made available upon study completion.

**Funding:** The study is funded by Programa de Apoio ao Desenvolvimento Institucional do Sistema Único de Saúde (PROADI-SUS, Brazil) (NUP 25000.014325/2021-33). LEM has a grant from FAPESP (18/24561-5) and from CNPq (311619/2019-3). The funders had and will not have a role in study design, data collection and analysis, decision to publish, or preparation of the manuscript.

**Competing interests:** The authors have declared that no competing interests exist.

**Abbreviations:** APOEε4, ε4 allele of the apolipoprotein E gene; βHCG, Human chorionic gonadotropin β; CT, Computerized tomography; DMC, Data Monitoring Committee; eCRF, Electronic case report form; ECU, Emergency care unit; ECU, Emergency care unit; EEG, Electroencephalogram; EQ VAS, EuroQol visual analogue scale; EQ-5D-3L, EuroQol three-level version; FDT, Five Digit Test; GCP, Good Clinical Practice; GCS, Glasgow Coma Scale; ICH, International Conference on Harmonization; ICU, Intensive care unit; mRS, modified Rankin Scale; PROADI-SUS, Programa de Apoio ao Desenvolvimento Institucional do Sistema Único de Saúde; PTE, Post-traumatic epilepsy; RAVLT, Rey Auditory Verbal Learning Test; REDCap, Research Electronic Data Capture; SAS, symptomatic acute seizures; TBI, Traumatic brain injury; WAIS III, Wechsler Intelligence Scale III.

## Discussion

We hypothesize that biperiden treatment will be effective to prevent or mitigate the development of post-traumatic epilepsy in TBI patients. Other health measures from this population also may benefit from treatment with biperiden.

## Trial registration

ClinicalTrials.gov, NCT04945213. Registered on June 30, 2021.

## Introduction

Traumatic brain injury (TBI) is one of the main leading causes of death and lifelong disability in the world. One of the most important neurological consequences following TBI is the development of post traumatic epilepsy (PTE), which is responsible for approximately 5% of all focal epilepsy etiologies in the general population [1]. TBI patients present a higher risk to develop epilepsy than the general population, with 10–20% cumulative probability in 5 years [2–4]. The type and severity of injury correlates with the likelihood for a person to develop epilepsy, varying from 2.9% for moderate TBI to 17–27% for those who had severe TBI [2, 5–7]. Importantly, risk of epilepsy reaches 50% for patients with penetrating head injury [8]. Besides the high incidence, the causative links between TBI and epilepsy are still not completely understood, and there is currently no clinically available treatment or strategy that influences the likelihood of PTE development after brain injury [9].

Preclinical studies showed that anticholinergic drugs have the potential to modify neural plastic processes hampering epileptogenesis in animal models of epilepsy [10]. Among these drugs, biperiden, a cholinergic antagonist acting in the muscarinic receptor, administered soon after brain injury, delayed the latency and decreased the incidence and intensity of spontaneous seizures after brain injury [11]. These results were recently corroborated by other groups [12].

As a subsequent step, the use of biperiden in the acute phase after brain injury showed to be safe in a small phase II clinical study, which also served to test the feasibility of an interventional and prospective trial [10]. Here, the present Phase III study was designed to evaluate the safety and efficacy of biperiden in preventing the development of PTE in adult patients, who have suffered moderate or severe TBI, in a larger multicenter prospective study. Other clinically relevant neurological outcomes due biperiden treatment will also be investigated. The aim of this study is to evaluate the efficacy and safety of biperiden in preventing the development of epilepsy in patients that suffered TBI.

## Materials and methods

It is a multicentric phase 3 randomized, double-blind clinical trial coordinated by Sociedade Beneficente de Senhoras, Hospital Sirio-Libanês (Brazil), throught a partnership with the Brazilian Ministry of Health and funded by the Programa de Apoio ao Desenvolvimento Institucional do Sistema Único de Saúde (PROADI-SUS, Brazil). The study was prospectively registered in ClinicalTrials.gov (NCT04945213). This reporting followed the recommendations of the Standard Protocol Items: Recommendations for Interventional Trials (SPIRIT Statement) [13] and the Template for Intervention Description and Replication (TIDieR-placebo) checklist [14].

Fig 1 presents the study plan including the schedule of enrolment, intervention, and assessments. A complete SPIRIT checklist for clinical trial can be found on the S1 Checklist.

## Ethical considerations

This study follows the principles of the Good Clinical Practice (GCP–ICH6 R2) in Randomized Controlled Trial (RCT). The whole protocol has been reviewed and approved by the Research Ethics Committee of Sociedade Beneficente de Senhoras, Hospital Sírio-Libanês (CAAE No. 39005920.8.1001.5461). Because the emergency nature of TBI, the Research Ethics Committee allowed to initiate the study protocol without informed consent of patients (or relatives), which must be provided until 48h.

An external Data Monitoring Committee (DMC) formed by three experienced clinic-researchers will provide independent assessment of the continuity of the study consiÞring the safety, validity, ethical and scientific merit of the trial.

## Participants

The study will enroll adults (18 to 75 years) of both sex who fulfill the following inclusion criteria: meet the moderate or severe TBI criteria according with the Glasgow Coma Scale (GCS) [15], in such a way that GCS is between 6 and 12 at the accident scene and/or between 3 (if sedated) and 12 at hospital admission; have computerized tomography (CT) scan with evidence of acute intraparenchymal hemorrhage and/or contusion; able to receive the first dose of treatment or placebo within 12 hours of brain injury, regardless of the accident cause. Participants, or their relatives in the event of patient´s unconsciousness, will have to provide written informed consent within 48 hours from the beginning of the protocol. Exclusion criteria are: current use of biperiden or antiseizure drugs; history of epilepsy, presence of other conditions that could increase risk factors for epilepsy development, such as perinatal injuries, meningitis, encephalitis, neoplasia, acute stroke, neurodegenerative diseases; any condition precluding the use of biperiden such as pregnancy, glaucoma, benign prostatic hyperplasia, atrioventricular block or cardiac arrhythmias; megacolon or mechanical obstruction of the gastrointestinal tract and inability to perform follow-up visits (e.g. homeless patients); refusal to participate in the study and current enrollment in other clinical trial. Alcohol intoxication or other psychotropic drug use at hospital admission will not lead to protocol exclusion.

We intend to recruit participants with acute TBI from at least ten hospitals located at different regions from Brazil (North, South, Southeast, Midwest and Northeast). After a first screening at the emergency care unit (ECU) for confirming eligibility criteria (which requires standard CT scan for acute TBI cases, electrocardiogram and pregnancy test for safety), those who fulfill all criteria will be included in the study. Because the emergency scenario and the injury severity, when the participant is not able to inform his medical history and is not accompanied by an appropriate informant, consent to participate and exclusion criteria might be confirmed within two days of protocol beginning. Participants will be prospectively monitored at their respective neurology/neurosurgery centers.

## Randomization and allocation concealment

Patients will be randomly assigned to receive placebo or biperiden using a random table generated by STATA® v17 software, in a 1:1 rate, in random permuted blocks and considering the recruiting centers as a criterion for stratification. The allocation concealment will be assured using the REDCap (Research Electronic Data Capture), a web-based software platform [16, 17].

**Study Plan**

Enrolment and Intervention (Days) — Follow up (Months)

| Timepoint after TBI | 1 | 2 | 3 | 4 | 5 | 6 | 7 | 8 | 9 | 10 | 1[a] | 3[b] | 6[b] | 9[b] | 12[b] | 18[b] | 24[b] |
|---|---|---|---|---|---|---|---|---|---|---|---|---|---|---|---|---|---|
| **ENROLMENT** | | | | | | | | | | | | | | | | | |
| Eligibility screening — CT, ECG, βHCG | x | | | | | | | | | | | | | | | | |
| Clinical history and demographic data | x | x | x | x | x | | | | | | | | | | | | |
| Informed consent | x | x | | | | | | | | | | | | | | | |
| **Allocation** | x | | | | | | | | | | | | | | | | |
| **INTERVENTION** — Study drugs iv infused | x | x | x | x | x | x | x | x | x | x | x | | | | | | |
| **ASSESSMENTS** | | | | | | | | | | | | | | | | | |
| *Primary Outcomes* — PTE incidence | | | | | | | x | x | | x | x | x | x | x | x | x | x |
| Serious adverse events occurrence | x | x | x | x | x | x | x | x | x | x | x | x | x | x | x | x | x |
| *Secondary Outcomes* — Clinical evolution, occurrence of SAS, APOEε4 | x | x | x | x | x | x | x | x | x | x | | | | | | | |
| Clinical and neurological assessment, mRS | | | | | | | | | | | x | x | x | x | x | x | x |
| EEG | | | | | | | | | | | x | x | x | x | x | x | x |
| *Quality of life* — EQ-5D-3L, EQ VAS, sociodemographic data | | | | | | | | | | | x | | x | | x | | x |
| *Neuropsychologic* — WAIS (Vocabulary, Block design, Digit Symbol-Coding, Symbol Search and Digit Span) Rey–Osterrieth complex figure, RAVLT, FDT | | | | | | | | | | | | | x | | | | x |
| *Other Outcomes* — Mortality incidence | x | x | x | x | x | x | x | x | x | x | x | x | x | x | x | x | x |
| Non-severe adverse events occurrence | x | x | x | x | x | x | x | x | x | x | x | x | x | x | x | x | x |

\*Standard protocol items recommendations for Interventional Trials (SPIRIT) diagram [13].

CT computerized tomography; ECG electrocardiogram, βHCG human chorionic gonadotropin β; PTE post-traumatic epilepsy; SAS symptomatic acute seizures; EEG electroencephalography; mRS modified Rankin scale; EQ-5D-3L EuroQol three-level; EQ VAS EuroQol visual analogue scale; WAIS Wechsler Intelligence Scale; RAVLT Rey Auditory Verbal Learning Test; FDT Five Digit Test; polymorphic ε4 allele of the apolipoprotein E gene.

Time deviations in days (d) allowed for follow up visits in months(m): a 1m ± 5d; b 3, 6, 9, 12, 18, 24m ± 15d.

**Fig 1. Biperiden study plan including the schedule of enrolment, intervention, and assessments**\*. \*Standard protocol items recommendations for Interventional Trials (SPIRIT) diagram [13]. CT computerized tomography; ECG electrocardiogram, βHCG human chorionic gonadotropin β; PTE post-traumatic epilepsy; SAS symptomatic acute seizures; EEG electroencephalography; mRS modified Rankin scale; EQ-5D-3L EuroQol three-level; EQ VAS EuroQol visual analogue scale; WAIS Wechsler Intelligence Scale; RAVLT Rey Auditory Verbal Learning Test; FDT Five Digit Test; polymorphic ε4 allele of the apolipoprotein E gene. Time deviations in days (d) allowed for follow up visits in months(m): a 1m ± 5d; b 3, 6, 9, 12, 18, 24m ± 15d.

## Blinding procedures

The intervention and the placebo will be identical in appearance (color, volume and packaging). Participants, staff and outcome assessor will be blinded to participant assignments.

## Interventions

Within 12 h after TBI, participants will receive intravenous (i.v.) infusions of 5 mg of biperiden (1 mL; Cinetol, Cristália, Brazil) diluted in 100 mL of 0.9% saline (treatment group) or 1 mL of sterile vehicle (sodium lactate, lactic acid, sodium hydroxide and water for injections) diluted in 100 mL of 0,9% saline (placebo group), every 6 hours for 10 days after TBI, until completing 40 total doses.

Patients will receive all standard of care treatment indicated for their clinical condition according to the hospital protocol and will not be deprived of any other required treatment, including all types of anti-seizure drugs, because of their participation in this study, regardless of the randomization group (placebo or biperiden) or the protocol period (intervention or follow up). Patients can be discontinued from the protocol if they remove the consent to participate; in any health conditions that health care providers, with sponsor agreement, believe that it is to the advantage of the patient not to comply with the procedures specified in this protocol; in case of screening error when including patients with history of epilepsy predicted in the study exclusion criteria (when this finding cannot occur at the time of recruitment).

During the hospitalization period, blood samples will be collected to analyze the genetic profile of patients regarding the presence of the polymorphic ε4 allele of the apolipoprotein E gene (APOEε4). Laboratory analyses will be performed by a central laboratory (DASA, São Paulo, Brazil), which is accredited by the College of American Pathologists (CAP, Northfield, IL, USA). Additional patients' information, such as occurrence of adverse events, sociodemographic data, occurrence of symptomatic acute seizures, unprovoked epileptic seizures starting 7 days after TBI and clinical evolution will be collected during the hospitalization period. Suspected and diagnosed severe adverse events will be reviewed and adjudicated by the Ethics Committee and the external Data Monitoring Committee.

## Follow-up

At hospital discharge, patients and their relatives will be asked to keep a diary of epileptic seizures and record all ictal events with detailed descriptions. Prospectively, patients will be followed up for two years, on periodic visits at 1, 3, 6, 9, 12, 18 and 24 months after TBI with a neurologist or neurosurgeon, in neurology centers, to assess the development of epileptic seizures. Specifically, seizure diary, clinical and neurological evaluation, which include the Portuguese version of the modified Rankin Scale (mRS) for neurologic disability [18], will be assessed. In case of prolonged hospitalization, follow up visits will occur at the hospital, as possible. Due to the COVID-19 pandemic, we anticipate performing some of these evaluations through telephone calls. Despite the limited clinical assessment promoted by this resource (e.g. loss of physical evaluation), it allows investigation over seizure occurrence.

Patients will be further monitored with electroencephalogram (EEG) at the same time points defined for clinical evaluation. EEG recordings will be analyzed by experienced neurophysiologists. Other personal aspects that might benefit from the treatment or might influence its results will also be evaluated on specific timepoints, as shown on Fig 1, which include: social data and quality of life assessed using the Portuguese version of EuroQol three-level version (EQ-5D-3L) and EuroQol visual analogue scale (EQ VAS) at 3, 6, 12 and 24 months after TBI; standard neuropsychologic tests, including test batteries (Vocabulary, Block design, Digit Symbol-Coding, Symbol Search and Digit Span) of the Wechsler Intelligence Scale III (WAIS III); Rey–Osterrieth complex figure; Rey Auditory Verbal Learning Test (RAVLT) and Five Digit Test (FDT) which will be applied by psychologists at 6 months and then 24 months after TBI.

Data will be collected in an electronic case report form (eCRF) using free web-based software platform REDCap. Data entry will be monitored by an independent researcher according to a predefined monitoring plan. Patient confidentiality will be ensured by using identification numbers. All records will be kept on file at the trial sites or the coordinating data center for 5 years. The frozen trial database file will be kept on file for 5 years.

## Outcomes

The primary and secondary outcomes as well as planned analysis are listed in Table 1.

As primary outcome, we will investigate the number of participants with unprovoked seizures, counted starting 7 days after TBI until the two years follow-up period. The presence of seizures will be clinically confirmed by a neurologist to indicate PTE. Also, as a primary outcome, the number of patients experiencing at least one serious adverse event in the two years follow-up period will be compared between placebo and biperiden groups. The ICH Guideline for Clinical Safety Data Management: Definitions and Standards for Expedited Reporting defines serious adverse event as: "A serious adverse event (experience) or reaction is any untoward medical occurrence that at any dose: results in death, is life-threatening (NOTE: The term "life-threatening" in the definition of "serious" refers to an event in which the patient was at risk of death at the time of the event; it does not refer to an event which hypothetically

**Table 1. Planned analysis for primary and secondary outcomes.**

| Description | Measurement | Timepoint (months) |
|---|---|---|
| **Primary Outcomes** | | |
| PTE incidence | Number of participants with unprovoked seizures | 24 |
| Serious adverse events occurrence | Number of participants experiencing at least one serious adverse event | 24 |
| **Secondary Outcomes** | | |
| PTE incidence | Number of participants with unprovoked seizures | 1, 3, 6, 9, 12, 18, 24 |
| mRS score | Number of participants categorized according to total score of the mRS scale | 1, 3, 6, 9, 12, 18, 24 |
| EEG pattern | Number of participants presenting epileptiform discharges and electrographic seizures | 1, 3, 6, 9, 12, 18, 24 |
| Neuropsychological Status | Number of participants categorized according to percentile ranks of neuropsychological tools | 6, 24 |
| Quality of life score | Number of participants categorized according to specific and total scores of quality of life scales | 3, 6, 12, 24 |
| **Other Outcomes** | | |
| Mortality incidence | Number of participants that evolved to death (all cause mortality) | 24 |
| Non-serious adverse events occurrence | Number of participants experiencing at least one non-serious adverse event | 1, 3, 6, 9, 12, 18, 24 |
| **Subgroup analysis** | Age, gender, region of lesion, occurrence of SAS, frequency and type of epileptic seizures, presence of APOEε4, educational level. | |

The data analysis is based on comparisons between biperiden x placebo groups considering the end of follow up or run-in periods. PTE post-traumatic epilepsy; EEG electroencephalography; mRS modified Rankin scale. SAS symptomatic acute seizures; APOEε4 polymorphic ε4 allele of the apolipoprotein E gene.

might have caused death if it were more severe); requires inpatient hospitalization or prolongation of existing hospitalization, results in persistent or significant disability/incapacity, or is a congenital anomaly/birth defect" [19].

For secondary outcomes, it will be considered the frequency and type of clinical seizures at each specific timepoint during the two years follow up. Also, the number of participants presenting epileptiform discharges and electrographic seizures on the electroencephalogram recording will also be analyzed. For the neuropsychological status participants will be categorized according to the specific percentile rank defined for each cited standard test. Five health-related quality of life domains (mobility, usual care, usual activities, pain/discomfort, anxiety/depression) will be covered and compared between groups according to specific and total scores.

Other outcomes include the mortality incidence and occurrence of non-serious adverse events during the two year protocol.

## Statistical analysis

The statistic plan followed the Guidelines for the Content of Statistical Analysis Plan in Clinical Trials [20].

Sample size calculation was performed based on previous studies [1, 2, 5–8, 10, 21] using the Pocock formula [22]. Assuming a difference in reduction of PTE of 15% between the placebo and biperiden groups at 24 months, type 1 error (alpha) of 0,05 and type 2 error (beta) of 0.10 and 20% dropout rate, 156 patients per group would be required to provide 90% power with 5% level of significance. According to epidemiological data provided by a pilot study [10], we estimated that the inclusion pace could be 3–5 patients/center/month leading to study completion within 42 months.

The hypothesis used in the analysis is that biperiden is superior to placebo. All patients who receive at least one dose of randomized study drug will be included into the safety analysis and all patients who receive at least one dose of a randomized study drug and who has an evaluable follow-up data will be included in intention-to-treat analysis. Missing data will be handled by chained-equation multiple imputation methods considering relevant baseline and outcome variables. A complete case analysis will be presented as a sensitivity analysis.

We plan to analyze the primary outcome using logistic regression, adjusting for age and sex at baseline. The result of this regression will be an odds ratio representing the relative effect of the intervention compared to the placebo. Secondary outcomes will be analyzed using appropriate generalized linear models, including logistic and linear models. Planned subgroups include type of epileptic seizures and the presence of the polymorphic ε4 allele of the apolipoprotein E gene (Table 1).

An interim analysis will be performed at 50% recruitment considering the primary outcomes. At the interim enrollment point, the DMC will consider as criteria for early interruption of the study p<0.001 for safety and p<0.0001 for efficacy analysis. The Pocock [22] stopping boundaries, will be used as a reference rather than as a rigid rule. In addition, the DMC will also consider the effect on other secondary outcomes, in the occurrence of unpredicted adverse events and external new evidence available during trial conduction.

A confidence level of 95% will be assumed for all analyses that will be conducted using Stata v17.

## Discussion

This is a multicentric interventional prospective study aiming to evaluate the efficacy and safety of biperiden in preventing the development of epilepsy in patients that suffered TBI, in a

double blind, randomized, placebo-controlled trial. If successful, it will be the first antiepileptic drug and will confirm the safety of the drug in patients with TBI, with the advantage of having low cost and good availability given it is a medication already in use in the clinical practice and which has been extensively used for another medical condition.

The process of epileptogenesis has been the subject of numerous studies and despite some promising results in laboratory settings no compound has yet advanced to the clinical practice [23]. Both clinical and laboratory evidence suggest crucial roles for the glutamatergic and the GABAergic systems in the expression and suppression of seizures and also in the development of the epileptic condition. The cholinergic system on the other hand, raises a limited interest from the clinical perspective (with some notable exceptions such as soman and sarin) in the epilepsy arena [24]. However, recently different strategies to modulate neuronal cholinergic activity have been shown to be effective at modifying the epileptogenic process in preclinical studies [10, 12]. Among various approaches, acute treatment with biperiden, showed to hamper the development of PTE in animal models of brain injury [11].

Here we move forward by developing a clinical trial, in which biperiden will be administered in brain injured patients as soon as possible after the traumatic brain injury and only for a period of 10 days. The emergency room settings as well as the critical patient conditions (often life threatening) in addition to the long-term period (2 years) to follow up patients, imposes practical and operational hurdles for the successful development of the study. A pilot study carried out under similar conditions [10] helped to outline the main difficulties and strategies to overcome these limitations, as discussed below.

Because treatment has to be administered soon after injury, and because severe trauma patients may be unconscious and not accompanied by a legal responsible person at the hospital admission, research ethics committees from all study centers allowed an extended deadline of 48h to obtain the informed consent to the study after beginning treatment, in order to guarantee effective enrollment rate. Also, because some aspects of the clinic history may not be readily available during enrollment, some inclusion and exclusion criteria will be checked over the initial course of treatment, which may result in high exclusion rate of patients. In that case, data from those patients will be considered only for safety analysis.

The dose of intravenous injection of biperiden is similar to the maximum recommended dose to initiate treatment of severe cases of parkinsonism [25] and also showed to be safe in critical TBI patients [10]. However, severely injured patients receive a variety of treatments and drug interventions and, in addition to the evolution of the clinical condition, all those are confounding factors that must be taken in account when registering adverse effects that might be caused by the biperiden treatment. We will focus the analysis of adverse effects on the symptoms already described for biperiden without underestimating additional unreported occurrences, mainly during treatment, and its correlation with other possible causes.

The long-term assessments in different areas (EEG, neuropsychology, etc) will be conduct by specialized staff (neurologists, neurophysiologists, trained psychologists). To decrease loss of data associated to absence of the patient in presential visits on specified time points, e.g because social restrictions due COVID-19 pandemic or other reasons, essential data related to clinical evaluation of seizures development will be collected by phone calls, as necessary. Different contact options, such as relatives phone numbers and e-mail addresses will be collected and only after many frustrated attempts of contact for different clinical assessments, patients will be considered lost in the follow up.

The present study will focus on patients with moderate to severe TBI, comprising the group with the highest risk to develop PTE. While the pathogenesis of epilepsy development after brain injury remains unclear, the search for earlier biomarkers and for new resources that might enable to identify patients who could benefit from prophylactic treatments are

additional highlights of this work. We also will investigate whether the anticholinergic actions of biperiden may also have some impact on other outcomes or sequelae after TBI. Accordingly, the present study will provide major information related to: (i) the proportion of patients who develop PTE in addition to those who die in the two-year follow up period after TBI; (ii) other aspects of the patients'neurological outcomes that may benefit from early treatment, such as degree of disability or dependence in the daily activities, general life quality and neuropsychological dimensions; and (iii) the incidence of different ApoE polymorphisms, more importantly of APOEε4 and its putative association to epileptogenesis and treatment effectiveness.

An additional positive aspect of this project is to use an already known drug, affordable and widely used to treat another neurological pathology, being part of the list of Essential Medicines developed by the World Health Organization which indicates medicines to be always available within general health systems in adequate amounts. In the event this drug is shown to be effective in reducing epilepsy after TBI in humans it would be readily available for use as an off-label indication. Moreover, it might also have the potential to be used to mitigate the development of epilepsy secondary to other pathologies, such as after stroke.

## Supporting information

**S1 Checklist. SPIRIT 2013 checklist: Biperiden trial for epilepsy prevention (BIPERIDEN).** (DOCX)

**S1 File.**
(DOCX)

**S2 File.**
(DOCX)

**S3 File.**
(DOCX)

**S4 File.**
(PDF)

## Acknowledgments

Debora Patricio Silva and Camila Santana Justo Cintra Sampaio for their support in the construction and elaboration of electronic Case Report Forms (CRFs) in the REDCap databases.

## Author Contributions

**Conceptualization:** Eliana Garzon, Luiz Eugênio Mello.

**Data curation:** Carla Cristina Gomes Pinheiro.

**Formal analysis:** Maira Licia Foresti, Eliana Garzon, Rafael Leite Pacheco, Rachel Riera, Luiz Eugênio Mello.

**Funding acquisition:** Maira Licia Foresti, Eliana Garzon, Luiz Eugênio Mello.

**Investigation:** Eliana Garzon.

**Methodology:** Maira Licia Foresti, Eliana Garzon, Carla Cristina Gomes Pinheiro, Rafael Leite Pacheco, Rachel Riera, Luiz Eugênio Mello.

**Project administration:** Maira Licia Foresti, Carla Cristina Gomes Pinheiro, Luiz Eugênio Mello.

**Resources:** Carla Cristina Gomes Pinheiro, Luiz Eugênio Mello.

**Supervision:** Maira Licia Foresti, Eliana Garzon, Carla Cristina Gomes Pinheiro, Luiz Eugênio Mello.

**Validation:** Maira Licia Foresti, Eliana Garzon, Carla Cristina Gomes Pinheiro.

**Writing – original draft:** Maira Licia Foresti, Eliana Garzon.

**Writing – review & editing:** Eliana Garzon, Carla Cristina Gomes Pinheiro, Rafael Leite Pacheco, Rachel Riera, Luiz Eugênio Mello.

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
