## [Decision Letter · Decision Letter 0]

24 Jun 2022

PONE-D-22-1613Biperiden for prevention of post-traumatic epilepsy: a protocol of a double-blinded placebo-controlled randomized clinical trial (BIPERIDEN trial)PLOS ONE

Dear Dr. Mello,

Thank you for submitting your manuscript to PLOS ONE. After careful consideration, we feel that it has merit but does not fully meet PLOS ONE’s publication criteria as it currently stands. Therefore, we invite you to submit a revised version of the manuscript that addresses the points raised during the review process.

We believe that the authors will easily answer the reviewers' comments, but I would also require some clarifications regarding the primary outcome:-I would like to be sure that any event (mortality, hospitalization, disability etc...) during a 2 year period after inclusion, will be regarded as positive for the primary outcome. -Is there rationale that an early a short prevention of epilepsy will affect re-hospitalisation for any cause (for example infection)-Would it be relevant to focus on death/rehospitalization related to neurologic issue / epilepsy?-As these events are the primary endpoint, please provide a definition of life-trheatening event, prolonged hospitalisation, disability, permanent damage-Please clarify why congenital birth/defect is in the primary endpoint because it looks more like a safety endpoint to me

We look forward to receiving your revised manuscript.

Kind regards,

Raphael Cinotti, MD, PhD

Academic Editor

PLOS ONE

Journal Requirements:

" ext-link-type="uri" xlink:type="simple">https://journals.plos.org/plosone/s/file?id=ba62/PLOSOne_formatting_sample_title_authors_affiliations.pdf"

Reviewers' comments:

Reviewer's Responses to Questions

**Comments to the Author**

1. Does the manuscript provide a valid rationale for the proposed study, with clearly identified and justified research questions?

Reviewer #1: Yes

Reviewer #2: Yes

2. Is the protocol technically sound and planned in a manner that will lead to a meaningful outcome and allow testing the stated hypotheses?

Reviewer #1: Yes

Reviewer #2: Yes

3. Is the methodology feasible and described in sufficient detail to allow the work to be replicable?

Reviewer #1: Yes

Reviewer #2: Yes

4. Have the authors described where all data underlying the findings will be made available when the study is complete?

Reviewer #1: Yes

Reviewer #2: Yes

5. Is the manuscript presented in an intelligible fashion and written in standard English?

Reviewer #1: Yes

Reviewer #2: Yes

6. Review Comments to the Author

You may also provide optional suggestions and comments to authors that they might find helpful in planning their study.

Reviewer #1: Dear authors,

Thank you for giving me the Opportunity to review your study protocol entitled "Biperiden for prevention of post-traumatic epilepsy: a protocol of a double-blinded placebo-controlled randomized clinical trial".

It is a well-written manuscript and I read it with great interests.

Here are my comments in order to improve the manuscript.

1- Line 211: sentence "... with the advantage of having a great degree of safety..."

I would nuance this sentence because safety is one of the stakes in this RCT. I suggest to replace this part of the sentence by a formulation, as an example, like "and will confirm the great safety of the drug"

2- Line 242: "...will be conduct by specialized personnel."

Does it mean that this staff is trained to administer questionnaires on Quality of Life?

Minor remark: maybe the word staff would be more appropriate instead of personnel (line 234)

Kind regards

Reviewer #2: This is an interesting study protocol that will address a significant medical issue. The scientific rationale is well described, I have just a few comments regarding the study design

- Ethic. Inclusion can be made without consent due to the clinical situation, but it is not clear if the consent of the patient will be sought after the first 48 hours. It seems important since a 2-year follow-up is planned

- Standard of care. A better description of the anti-epileptic treatment of the control group is required: are preventive treatments completely forbidden? In case of prolonged sedation in comatose patients, are benzodiazepines allowed?

- Intervention period. A description of the treatment of epilepsy in the case of cloni during the first 6 days period of treatment is needed.

- End of treatment. Can the treatment be stopped in patients with epilepsy and not yet controlled ?

- Follow-up. In the case of patients with difficult-to-treat intracranial hypertension, or protracted comatose states, what is the policy to rule out a status epilepticus.

7. PLOS authors have the option to publish the peer review history of their article (what does this mean?). If published, this will include your full peer review and any attached files.

Reviewer #1: **Yes: **Yoann Launey

Reviewer #2: **Yes: **Roquilly

---

## [Author Response · Author response to Decision Letter 0]

4 Jul 2022

Response to Reviewers

 Editor Comments 

Thank you for submitting your manuscript to PLOS ONE. After careful consideration, we feel that it has merit but does not fully meet PLOS ONE’s publication criteria as it currently stands. Therefore, we invite you to submit a revised version of the manuscript that addresses the points raised during the review process. 

Response: We appreciate the time and consideration given to our manuscript. The current version of the manuscript has been revised. We considered all points raised by the reviewers and editor and believe to have made the necessary changes to address their comments and suggestions, as specified bellow. 

We believe that the authors will easily answer the reviewers' comments, but I would also require some clarifications regarding the primary outcome.

1- I would like to be sure that any event (mortality, hospitalization, disability etc...) during a 2 year period after inclusion, will be regarded as positive for the primary outcome. 

Response: Indeed, we will consider all above mentioned aspects, in accordance with the definition of serious adverse event of the ICH guideline, as an outcome in our study. As described in the manuscript, “the number of patients experiencing at least one serious adverse event will be compared between placebo and biperiden groups”. To clarify we added the reference and definition of serious adverse event according to the ICH Guideline for Clinical Safety Data Management: Definitions and Standards for Expedited Reporting (1994): “A serious adverse event (experience) or reaction is any untoward medical occurrence that at any dose: results in death, is life-threatening (NOTE: The term "life-threatening" in the definition of "serious" refers to an event in which the patient was at risk of death at the time of the event; it does not refer to an event which hypothetically might have caused death if it were more severe); requires inpatient hospitalization or prolongation of existing hospitalization, results in persistent or significant disability/incapacity, or is a congenital anomaly/birth defect.”. In addition, to more clearly indicate the time period being analyzed in our study, we changed “at 24 months” by “in the two years follow-up period”. 

2 -Is there rationale that an early a short prevention of epilepsy will affect re-hospitalization for any cause (for example infection). 

Response: As answered in question 1, we will consider all items in the definition of serious adverse event of the ICH guideline as an outcome in our study. On hypothetical grounds a prolonged epileptic seizure, may lead to re-hospitalization of patients. There is no evidence that biperiden, may affect the occurrence of immediate seizures that may take place after traumatic brain injury. There is also no evidence that we know of an indirect action in another physiological aspect, that may alter the occurrence of re-hospitalizations, or even more so, whether or not administration of biperiden may alter the occurrence of other serious adverse events. Therefore, both for safety and efficacy, we will consider all serious adverse events listed in this study, for completeness of the investigation over the effects of biperiden treatment in patients with traumatic brain injury. 

3 -Would it be relevant to focus on death/rehospitalization related to neurologic issue / epilepsy?

Response: The objective of the intervention is to evaluate biperiden as an anti-epileptogenic drug, which may be possible over the two years follow-up after TBI. We will also assess any complications or re-hospitalizations that patients have. In this sense, clinical or neurological conditions will be considered. Please, see responses for questions 1 and 2 for rationale.

4 -As these events are the primary endpoint, please provide a definition of life-threatening event, prolonged hospitalization, disability, permanent damage

Response: Please, see response for question 1.

5 -Please clarify why congenital birth/defect is in the primary endpoint because it looks more like a safety endpoint to me.

Response: Please, see responses for questions 1 and 2.

Reviewers Comments 

Reviewer #1

Dear authors,

Thank you for giving me the Opportunity to review your study protocol entitled "Biperiden for prevention of post-traumatic epilepsy: a protocol of a double-blinded placebo-controlled randomized clinical trial".

It is a well-written manuscript and I read it with great interests.

Here are my comments in order to improve the manuscript.

Response: We appreciate the positive evaluation of the reviewer. In agreement with the reviewer, we accepted all suggestions, as specified bellow. 

1- Line 211: sentence "... with the advantage of having a great degree of safety..."

I would nuance this sentence because safety is one of the stakes in this RCT. I suggest to replace this part of the sentence by a formulation, as an example, like "and will confirm the great safety of the drug"

Response: The sentence has been replaced as suggested.

2- Line 242: "...will be conduct by specialized personnel."

Does it mean that this staff is trained to administer questionnaires on Quality of Life?

Response: This means that long-term assessments in different areas such as EEG, neuropsychology will be conduct by specialized staff such as neurologists, neurophysiologists and trained psychologists. This information was added in the revised version of the manuscript.

Minor remark: maybe the word staff would be more appropriate instead of personnel (line 234)

Response: In agreement with the reviewer suggestion, we have replaced the word personnel by staff.

Reviewer #2 

This is an interesting study protocol that will address a significant medical issue. The scientific rationale is well described, I have just a few comments regarding the study design

Response: We appreciate the positive evaluation of the reviewer.

1- Ethic. Inclusion can be made without consent due to the clinical situation, but it is not clear if the consent of the patient will be sought after the first 48 hours. It seems important since a 2-year follow-up is planned

Response: We appreciate the reviewer comments. In fact, after 48 hours, subsequent doses would only be applied after consent from the patient (or relatives). This information was added in the manuscript. Besides, as pointed in the item “Intervention”: Patients will be discontinued from the protocol in the event they remove the consent to participate.

2- Standard of care. A better description of the anti-epileptic treatment of the control group is required: are preventive treatments completely forbidden? In case of prolonged sedation in comatose patients, are benzodiazepines allowed?

Response: As stated in the item Intervention, “Patients will receive all medical treatment indicated for their clinical condition and will not be deprived of any other required treatment because of their participation in this study”. To further clarify, we have changed the sentence as follows: “Patients will receive all standard of care treatment indicated for their clinical condition according to the hospital protocol and will not be deprived of any required treatment, including all types of anti-seizure drugs, because their participations in this study, regardless the randomization group (placebo or biperiden) or the protocol period (intervention or follow up)”. 

3- Intervention period. A description of the treatment of epilepsy in the case of cloni during the first 6 days period of treatment is needed.

Response: The epileptic seizures that occur until the seventh day after TBI will be labeled as acute symptomatic seizures. As stated above, patients will receive all standard of care treatment indicated for their clinical condition according to the hospital protocol and will not be deprived of any required treatment, including all types of anti-seizure drugs, because their participations in this study, regardless the randomization group (placebo or biperiden) or the protocol period (intervention or follow up). 

4- End of treatment. Can the treatment be stopped in patients with epilepsy and not yet controlled ?

Response: History of epilepsy and presence of other conditions that could increase risk factors for epilepsy development are part of exclusion criteria to participate in the study. Patients developing post-traumatic epilepsy will not be deprived of any required treatment, as stated in response 2. 

5- Follow-up. In the case of patients with difficult-to-treat intracranial hypertension, or protracted comatose states, what is the policy to rule out a status epilepticus.

Response: As previously mentioned, the intervention with biperiden/placebo will not change the standard treatment of any patient. All procedures will be taken according to the needs of each patient. In the exposed situation, electroencephalogram may be requested to diagnose or exclude status epilepticus. Once the diagnosis of status epilepticus is confirmed, the appropriate treatment with benzodiazepines, anti-seizure drugs or anesthetics will be carried out according to each case and precise indication.

---

## [Decision Letter · Decision Letter 1]

11 Aug 2022

Biperiden for prevention of post-traumatic epilepsy: a protocol of a double-blinded placebo-controlled randomized clinical trial (BIPERIDEN trial)

PONE-D-22-16132R1

Dear Dr. Mello,

We’re pleased to inform you that your manuscript has been judged scientifically suitable for publication and will be formally accepted for publication once it meets all outstanding technical requirements.

Kind regards,

Hugh Cowley

Staff Editor

PLOS ONE

Additional Editor Comments (optional):

Reviewers' comments:

Reviewer's Responses to Questions

**Comments to the Author**

1. Does the manuscript provide a valid rationale for the proposed study, with clearly identified and justified research questions?

Reviewer #1: Yes

Reviewer #2: Yes

2. Is the protocol technically sound and planned in a manner that will lead to a meaningful outcome and allow testing the stated hypotheses?

Reviewer #1: Yes

Reviewer #2: Yes

3. Is the methodology feasible and described in sufficient detail to allow the work to be replicable?

Reviewer #1: Yes

Reviewer #2: Yes

4. Have the authors described where all data underlying the findings will be made available when the study is complete?

Reviewer #1: Yes

Reviewer #2: Yes

5. Is the manuscript presented in an intelligible fashion and written in standard English?

Reviewer #1: Yes

Reviewer #2: Yes

6. Review Comments to the Author

You may also provide optional suggestions and comments to authors that they might find helpful in planning their study.

Reviewer #1: Thank you for having adressed the reviewer's comments. I have no supplementary coments and think the manuscript can go further in the publication process

Reviewer #2: The authors have appropriately responded to my comments and I congratulate them for promoting such a study

7. PLOS authors have the option to publish the peer review history of their article (what does this mean?). If published, this will include your full peer review and any attached files.

Reviewer #1: **Yes: **Yoann Launey, MD, PhD

Reviewer #2: **Yes: **Antoine Roquilly

---

## [Editor Report · Acceptance letter]

31 Aug 2022

PONE-D-22-16132R1 

Biperiden for prevention of post-traumatic epilepsy: a protocol of a double-blinded placebo-controlled randomized clinical trial (BIPERIDEN trial). 

Dear Dr. Mello:

I'm pleased to inform you that your manuscript has been deemed suitable for publication in PLOS ONE. Congratulations! Your manuscript is now with our production department. 

Kind regards, 

on behalf of

Mr Hugh Cowley 

Staff Editor

PLOS ONE